# Market-based instruments to fund nature-based solutions for flood risk management can disproportionately benefit affluent areas

Bartholomew Hill [1] ✉, Tim Marjoribanks [2], Harriet Moore[3], Lee Bosher [4] & Mark Gussy[3]

Market-based instruments, including competitive tenders, are central to funding global environmental restoration and management projects. Recently, tenders have been utilised to fund Nature-based Solutions schemes for Natural Flood Management, with the explicit purpose of achieving co-benefits; flood management *and* reducing inequities. While multiple studies consider the efficacy of Nature-based Solutions for tackling inequities, no prior research has quantified whether the resource allocation for these projects has been conducted equitably. We analyse two national natural flood management programmes funded through competitive tenders in England to explore *who benefits* by considering the characteristics of projects, including socio-economic, geographical (e.g. rurality) and flood risk dynamics. Our results suggest that inequity occurs at both the application and funding stages of Nature-based Solutions projects for flood risk management. This reflects wider international challenges of using market-based instruments for environmental resource allocation. Competitive tenders have the potential to undermine the equitable benefits of Nature-based Solutions.

Internationally, a common approach to resource allocation for environmental restoration and management is through the use of market-based instruments[1–3]. Competitive tender funding is a prominent example and has an international legacy through national programmes such as the Bush-Tender scheme in Australia, the Countryside Stewardship Scheme in the UK and the Conservation Reserve Program in the US[4–7]. 'Competitive tender' allows open bidding for project funding where the most economically competitive (e.g. greatest cost-benefit ratio or lowest cost) tender option is selected. This approach is often utilised in situations of financial constraint to determine the most cost-effective allocation of public funding for greatest benefits, such as the contracting of Danish road and park services[8] and construction projects across the UK, US and New Zealand, although these applications of tender often encounter risks of poor quality assets[9,10]. In recent years, competitive tender funding approaches have been applied to Nature-based Solutions (NbS). Globally, the public sector provides 82% of all finance for NbS projects[11]. Therefore, competitive funding schemes significantly determine how and where NbS are implemented.

NbS are defined as 'actions to manage and restore ecosystems to address societal challenges' (adapted from IUCN[12]) and are becoming increasingly prominent as an approach to climate change adaptation, with many studies seeking to quantify their efficacy for tackling hydro-meteorological hazards[13]. In the UK and European context, NbS aimed at tackling flood risk is often referred to as Natural Flood Management (NFM)[14]. Both NFM and NbS, more broadly, have become a popular approach as they have the potential to reduce inequity through their co-benefits by addressing social and well-being problems in an affordable manner[12,15,16]. Inequity in this context refers to the unfair and avoidable disparities between communities, as well as the unfair or unequal distribution of funding for NbS/NFM projects. Given known associations between well-being and access to green spaces like restored wetlands, funding inequity extends beyond the implications for flood protection. Additionally, reducing inequity associated with NFM/NbS implementation also has the potential for addressing unequal social mobility through facilitating the knowledge transfer for the development of soft and technical skills[17]. Realising this potential depends on two elements: ensuring equitable benefits for all people within regions and communities where NbS is implemented, and the equitable allocation of resources for implementing NbS between different regions and communities. Several studies have

¹School of Natural Sciences, University of Lincoln, Lincoln, UK. ²School of Architecture, Building, and Civil Engineering, Loughborough University, Loughborough, UK. ³Lincoln Institute for Rural and Coastal Health, University of Lincoln, Lincoln, UK. ⁴School of Business, University of Leicester, Leicester, UK. ✉e-mail: BHill@lincoln.ac.uk

addressed the former[18–20]; however, there is a distinct lack of studies that assess whether NbS funding resources have been equitably distributed, despite wider critiques in other disciplines around the equity of restoration and conservation project funding[15,21,22].

Competitive grant funding schemes may, in fact, be counter-productive to the ambitions of NbS due to their potential to exacerbate rather than reduce inequities, yet their use remains widespread internationally. Competitive grants for NbS require skilfully crafted applications with no guarantee of funding, thus potentially benefitting applicants that have both greater skills and capacity to prepare successful applications[23,24]. Further, competitive grants are typically oriented towards maximising economic efficiency in the pursuit of targeted environmental outcomes[25], and the deliverability of a primary benefit (e.g. flood risk) at the lowest economic cost[26], rather than provision of multiple benefits equitably[27–29]. Hence, there is a likelihood that the funding allocation process leads to an exacerbation of inequities.

Given the challenges associated with competitive funding schemes for NbS, this article explores the competitive funding allocation for NbS using an example from the UK. We analyse the implementation of two recent UK Government (Department for Environment, Food & Rural Affairs, DEFRA) competitively funded NFM schemes in England, to provide statistical evidence of the extent to which they distribute resources for natural flood risk management, and associated co-benefits, equally. The first programme provided £15 million of investment into pilot NFM schemes, with £1 million of this going to support 34 community projects, between 2017 and 2021[30]. The success of this programme led to a further £25 million made available to fund another 40 projects for improving flood resilience in 2023[31].

To assess how equitable the allocation of funding for NFM, within England, has been, we review how these national government funding programmes for NFM have been allocated geospatially, with consideration to deprivation, rurality and flood risk. To measure deprivation, rurality and flood risk; we use the UK Indices of Deprivation (IMD), an index used for assessing socioeconomic deprivation for small areas (~1500 people) where low deciles 1–4 represent less affluent areas, and 5–10 represent more affluent areas[32]; the UK, Rural Urban Classification (RUC), a categorisation of rurality for small areas[33]; and the UK, Flood and Social Vulnerability (FSV) Index, an assessment of flood risk for small areas[34]. Through analysing the distribution of funding across these metrics, we aim to identify potential biases in how funds are allocated.

Our analysis joins geolocational data from applications and successful projects across the NFM funding programmes in 2017 and 2023 to the deprivation, rurality and flood risk data, using the funding programme guidance to determine the characteristics of benefitting communities. We use a Monte Carlo sampling approach to assess the statistical representativeness of both applications and funded projects in terms of deprivation, rurality and flood risk. To our knowledge, this is the first quantitative assessment of the potential inequity of NFM/NbS resource allocation through competitive grants and serves as a cautionary example of potential inequities that can inform the implementation of NbS internationally.

## Results
### Representativeness of funded NFM sites
There is statistically significant bias ($p < 0.05$) against lower (more deprived) IMD deciles in allocation of funded projects compared to the national IMD distribution in terms of either the mean or minimum decile for all 2023 projects and the 2017 urban projects (Fig. 1). For the rural 2017 projects there is no significant bias ($p \geq 0.05$) for any test statistic.

Urban projects have a statistically significantly high mean IMD decile in the 2017 and 2023 programmes ($p_A = 0.019$ for both), due to significant over-representation of IMD deciles 7–10 ($p_D = 0.034$ and $p_D = 0.032$ respectively) combined with under-representation of deciles 1–4. For the 2023 urban projects, this under-representation is also statistically significant ($p_C = 0.043$). For the 2017 urban projects, despite a significantly high

minimum decile ($p_B = 0.014$)—deciles 1 and 2 are absent from the funded projects—the percentage of sites in IMD deciles 1–4 is not significantly low ($p_C = 0.20$), due to a relatively high proportion of projects in deciles 3 and 4.

The mean IMD decile for the rural projects is not significantly greater than the national distribution for either programme ($p_A = 0.60$ for 2017 and $p_A = 0.085$ for 2023). However, in both cases, there is under-representation of low deciles, which is statistically significant for the 2023 projects, both in terms of the minimum decile (5, $p_B = 0.015$) and the percentage in deciles 1–4 ($p_C = 0.015$).

### Representativeness throughout the funding process
For rural locations, the national distribution (Fig. 2a) shows a low proportion of areas in deciles 1–4 and a higher proportion in deciles 5–10, in comparison with the relatively flat profile in urban areas, justifying the disaggregation of the analyses. The distributions of the applications are statistically significantly different from the national distributions for the 2023 programme across both rural, $\chi^2(8, N = 149) = 12.94$, $p = 0.049$, and urban locations, $\chi^2(9, N = 98) = 22.00$, $p = 0.0089$. For the 2017 programmes, neither the rural locations, $\chi^2(7, N = 102) = 12.94$, $p = 0.074$, nor the urban, $\chi^2(9, N = 69) = 10.50$, $p = 0.3$, have significantly different distributions at the 0.05 level.

The rural NFM applications (Fig. 2a) under-represent IMD deciles 1–3 compared to the national distribution in both 2017 and 2023. Middle deciles (4–7) are consistently over-represented, with some evidence of under-representation at higher deciles (8–10). At the assessment (funding allocation) stage, deciles 1–4 are under-represented compared to applications for both 2017 and 2023, while middle deciles are generally further over-represented with some exceptions. Higher deciles show a mixed picture: In 2017, deciles 8 and 10 are under-represented, but decile 9 is over-represented. In 2023, all deciles above 6 were over-represented in the funded projects compared to the applications.

For urban locations (Fig. 2b), the applications are more representative of the national distribution across both programmes, with some isolated under-representation (e.g. deciles 1 and 5 in 2017, deciles 1 and 3 in 2023) and over-representation in decile 9. The anomalies in deciles 9 are consistently statistically significant (i.e. $p_9 = 0.022$ in 2017, $p_9 = 0.008$ in 2023). The patterns at the assessment stage for urban locations are distinct between the 2 years. In 2017, the lowest deciles (1–2) were under-represented, whereas in 2023, the lowest deciles 1–3 were all representative of applications. Deciles 8 and 9 are over-represented in the funded projects compared to the applications across both programmes.

### Potential influence of relative flood risk and existing protection
The national distribution of flood risk exposure (Fig. 3) is relatively even across IMD deciles, with 51–61% of Lower Super Output Areas (LSOAs) having less than 10% population exposure to flood risk, 32–41% having 10–50% flood risk exposure and 2–8% having >50% of the population exposed to flood risk. The low (<10%) and medium (10–50%) flood risk exposure distribution shows no clear trend with IMD decile, whereas there is a decrease in high flood risk exposure (>50%) areas in deciles 7–10. These data show that variability in underlying flood risk between deciles does not explain the trends shown in Figs. 1 and 2. Similarly, considering existing flood protection (Table 1), the IMD deciles with the largest deviation from the national average for percentage protected are deciles 1 and 10 (both low) and 3–5 (all high). There is no consistent trend between these two sets of deciles and funding success observable in Fig. 2, showing that existing protection does not explain the observed IMD trends.

Overall, the distribution of flood risk within applications is significantly different from the national distribution for 2017 ($\chi^2(2, N = 170) = 6.65$, $p = 0.036$) and 2023 ($\chi^2(2, N = 247) = 17.44$, $p = 0.0002$), with applications preferentially from medium flood-risk areas. However, there is substantial variability between deciles. For example, in the 2017 applications, low-risk exposure areas (<10% exposure) were substantially over-represented in applications within decile 4. In all but two deciles, funded projects slightly under-represent high flood risk exposure areas, However, the small sample

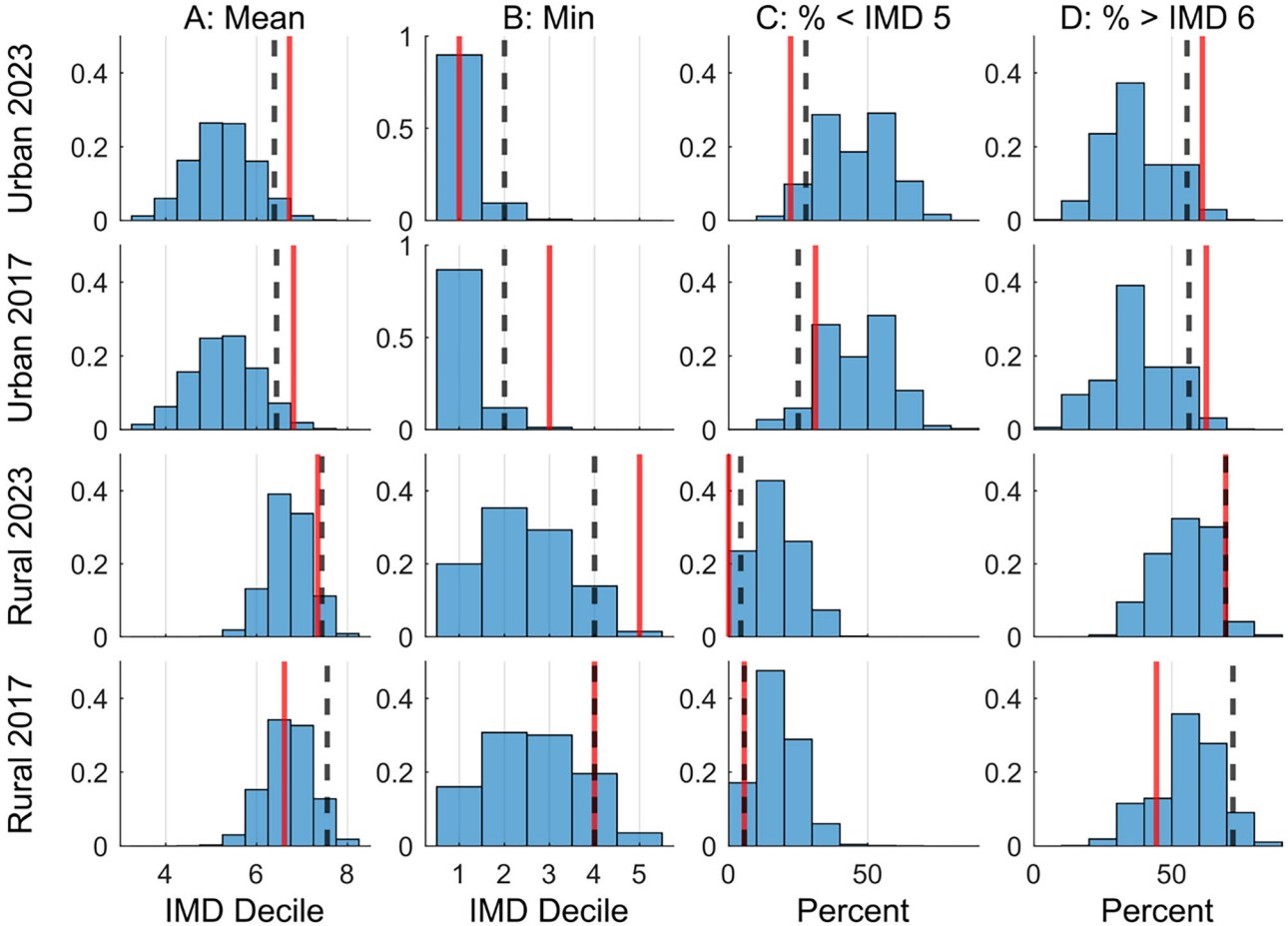

**Fig. 1 | A statistical analysis of how the IMD distribution of funded NFM projects for both rural and urban settings in 2017 and 2021 programmes compare to the national IMD distribution.** (First column) Compares the distribution of the mean IMD decile of funded 2017 and 2021 to the national distribution of IMD deciles, (second column) Compares the distribution of the minimum IMD decile of funded 2017 and 2021 to the national distribution of IMD deciles, (third column) compares the percentage of the lowest, and (fourth columb) compares the highest four IMD deciles across Monte Carlo samples taken from the national IMD distribution (M = 1,000,000). Black dashed lines indicate the one-sided 5th or 95th percentiles of the Monte Carlo samples, and red lines indicate the corresponding statistics from the NFM project data.

sizes within certain deciles can distort the results and overall funded projects were representative of the underlying national distribution of flood risk for 2017 ($\chi^2(2, N = 33) = 4.9$, $p = 0.086$) and 2023 ($\chi^2(2, N = 41) = 2.52$, $p = 0.28$).

Similarly, there is no significant bias in funding decisions towards either protected or unprotected LSOAs for 2017 ($\chi^2(1, N = 33) = 0.71$, $p = 0.4$) or 2023 ($\chi^2(1, N = 41) = 2.04$, $p = 0.15$) compared to the applications submitted. The applications themselves do over-represent LSOAs with existing protection (Table 1) compared to the national distribution, which is significant for the 2023 programme ($\chi^2(1, N = 247) = 6.30$, $p = 0.012$). This is expected as, at the national scale, protected LSOAs are inherently more likely to have flood risk than unprotected areas.

## Discussion

In principle, environmental management can promote social ecological justice, supporting national progress towards achieving the global sustainable development goals. In practice, programmes often encounter unforeseen challenges resulting in unintended consequences and missed opportunities for multi-benefits. It is not unusual for new initiatives like NbS to experience 'teething problems' with implementation, including funding mechanisms[35]. At this timely juncture, our analysis highlights key observations with feasible avenues for circumventing common shortcomings.

Firstly, our analysis of flagship UK NFM programmes demonstrates that *of all submitted applications for funding*, projects are disproportionately

funded in areas characterised by the highest IMD deciles, reflecting larger natural flood protection investment in more affluent areas. This suggests a lack of equity in the way that applications are *assessed* that is not explained by underlying trends in rural-urban dynamics, flood risk or existing flood protection; applications for projects in more deprived areas are less likely to be successful compared to applications for projects in less deprived areas. As a result, deprived areas with high flood risk receive less funding for NbS projects than their less deprived counterparts. Thus, while NbS are often promoted as an equitable approach to climate adaptation[18,36], the distribution of NbS projects does not reflect this ambition. Rather, competitively funded NFM grant programmes may preferentially benefit less deprived areas, contrary to DEFRAs Flood and Coastal Erosion Risk Management (FCERM) budget, which reports a weighting and prioritisation of funding for more deprived communities[37]. Given national[38] and international[39–41] commitments to substantially expand NbS programmes, we suggest that funding opportunities should be considered beyond current approaches to competitive tendering, which are widely critiqued for excluding local and marginalised groups, promoting short-term solutions and prioritising donor agendas over local needs[21,22,42]. Alternative approaches might include stratifying the tender process to ensure equitable opportunity for communities in more deprived and less deprived areas, towards reducing persistent 'green' inequalities[43,44], and promoting equitable flood protection through equitable fund allocations, as well as profound wider benefits to the community, such as soft and technical skills development, community empowerment and social mobility[45].

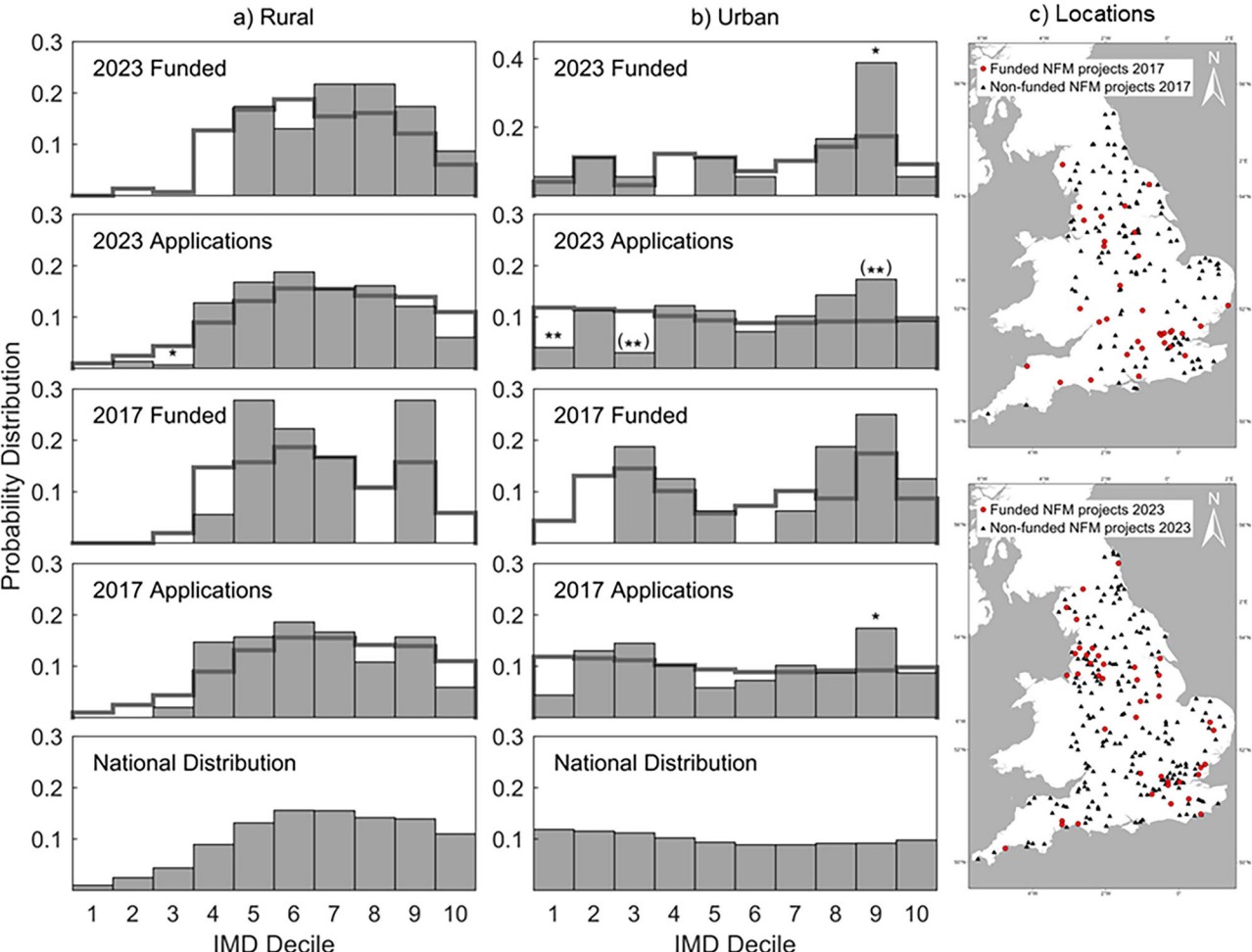

**Fig. 2 | A comparison of the IMD distribution at Lower Super Output Area (LSOA) scale for the national, applications and funded projects datasets. a** The IMD distribution for national, applications and funded NFM projects for 2017 and 2021 in Rural LSOAs. **b** The IMD distribution for national, applications and funded NFM projects for 2017 and 2021 in Urban LSOAs. **c** Two maps highlighting the spatial distribution of non-funded applications (black triangles), and funded applications (red circles), for both the 2017 and 2021 EA NFM funding programmes.

Shadow staircase plots show the underlying distribution from which each sample was drawn, i.e. for applications this shows the national distribution, while for funded projects, the staircase plots show the distribution of applications. Deciles with statistically significant difference compared to the underlying distribution are labelled single asterisk (*$p < 0.05$) or double asterisks (**$p < 0.01$). Deciles for which the statistical significance depends on the outputs of the project clustering algorithm (see 'Methods') are shown in parentheses.

Our second observation is that, in addition to inequity related to the assessment of submitted applications, inequity also occurs *at the application stage*. The most statistically significant anomalies were found at this stage. Figure 2 demonstrates that fewer applications were submitted to both programmes for projects in more deprived areas compared to less deprived areas, suggesting barriers to application. As a result, NbS funding seems to be concentrated within rural and urban communities that are more likely to possess the local expertise and capacity to undertake the complex application processes necessary for competitive tenders. Indeed, recent critiques of the competitive tender process/NFM projects highlight challenges around engagement with community, scepticism of the efficacy of NFM, lack of local expertise and capacity, and complexity within the application process (short time frames, skilful crafting, lack of flexibility, etc.)[38,46–49].

Reflecting on the under-representation of less affluent areas in both the funding decisions and application stages, this may stem from challenges of evidencing cost-benefit, and the perceived deliverability of the project[47,50]. Funding decisions are strongly influenced by the targeted assessment criteria rather than the multi-faceted equity-driven objectives that define NbS[25,26,47]. Quantifying benefits (particularly multi-benefits) and demonstrating capacity to deliver, monitor and evaluate complex projects requires capacity, resources and skills that some communities are more likely to have

than others, such as communities with a successful track record of project management. It is probable that difficulties with preparing compelling applications, including quantifying or monetising co-benefits, mean that funded projects are skewed toward those regions and communities that are already better-off, amplifying existing persistent inequities[27–29]. We suggest that funders should consider providing support for less experienced, under-resourced communities to guide the application process to reduce the risk of 'funding legacies' undermining the true potential of NbS to reduce rather than reinforce existing inequities and power structures that benefit the few and not the many[51].

Furthermore, in both the application and funding stages, we found no relationship between flood risk severity and NFM project siting, as well as fewer high flood risk areas within more affluent deciles, and so we can dismiss this as a potential explanation for the targeting of NFM within certain areas. This finding reflects NFM's suitability to tackle smaller scale 'nuisance' flooding rather than areas of very high flood risk[52]. However, there may also be some institutional bias towards affluent regions with localised flood risk. As NFM becomes more mainstreamed, it raises the question of whether it should be compared to hard infrastructure and FCERM, or whether it should be seen as complementary or implemented for alternative means, such as the associated wider benefits.

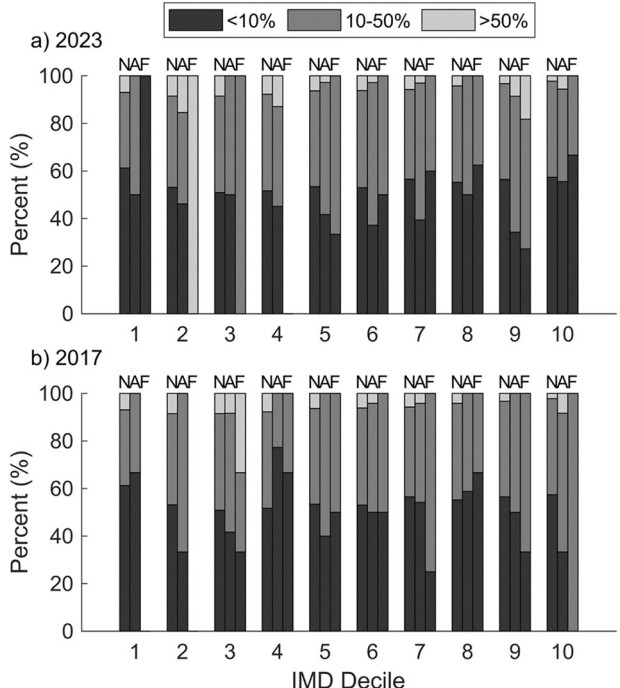

**Fig. 3 | A comparison of flood risk exposure in brackets of low flood risk (<10% LSOA coverage), moderate flood risk (10-50% LSOA coverage), and high flood risk (>50% LSOA coverage) against the IMD deciles for national, applications, and funded distributions.** Distribution of population percentage flood risk exposure at the LSOA level across IMD deciles for the **a** 2023 and **b** 2017 NFM programmes. The figure includes the national distribution across all LSOAs (N), the distribution across all application LSOAs (A) and all funded project LSOAs (N).

**Table 1 | Percentage of LSOAs with flood protection, by IMD decile for each stage of the 2017 and 2023 NFM programmes, compared to the national distribution**

| IMD decile | Percentage of LSOAs with existing flood protection | | | | |
| --- | --- | --- | --- | --- | --- |
| | National | 2017 | | 2023 | |
| | | Applications | Funded | Applications | Funded |
| 1 | 6.6 | 0 | 0 | 0 | 0 |
| 2 | 9.8 | 11.1 | 0 | 15.4 | 100 |
| 3 | 10.4 | 0 | 0 | 0 | 0 |
| 4 | 10.4 | 4.5 | 0 | 22.6 | 0 |
| 5 | 10.4 | 10 | 0 | 13.9 | 16.7 |
| 6 | 9.6 | 8.3 | 0 | 14.3 | 0 |
| 7 | 9.1 | 8.3 | 0 | 9.1 | 0 |
| 8 | 7.2 | 17.7 | 33.3 | 10.5 | 0 |
| 9 | 7.1 | 22.2 | 12.5 | 17.1 | 36.4 |
| 10 | 4.4 | 8.3 | 0 | 0 | 0 |
| Total | 8.5 | 10.6 | 6.1 | 13.0 | 17.1 |

Our final observation is about rural and urban differences in submitted and funded applications for NFM. Applications and projects were more prevalent within rural areas, which may indicate that NFM is typically viewed as a rural solution. However, within rural areas, 65 of the 210 unsuccessful project applications (31.0%) and 18 of the 41 projects funded (43.9%) across both programmes were located within either National Parks or Areas of Outstanding Natural Beauty, which together account for 24–28% of England's rural land area. A likely reason for the high representation of

NFM in rural areas is the associated ease of implementation on publicly or institutionally owned land[53]. The over-representation of these areas, which increases between the application and funding stages, shows that NFM was not delivered equitably across all rural areas. Instead, areas that are more pristine or with high natural value received more funding, while more degraded sites are less well represented in applications and funded projects[54]. Furthermore, our results suggest that applications from deprived urban areas were more likely to be retained and funded following the assessment stage than deprived rural areas, despite no stipulations in assessment criteria related to rurality or urbanity. For example, of projects funded in rural areas, none were established in areas categorised in the lowest 3 deciles. By comparison, projects funded in urban areas include a small number in areas categorised in the lowest deciles, suggesting some enabling factors within urban communities that improve the funding potential for these projects. One possibility is that higher rates of delivered projects in deprived urban areas relate to the characteristics of the applications themselves.

In summary, our results reflect two government-funded NFM programmes. While the percentage of publicly versus privately funded NFM projects in the UK is unknown, many researchers highlight that private funding for public goods and climate adaptation is often an exception, and not the rule[55], and it is commonly known that most environmental projects, including NFM in river basins and coastal areas, are government-funded[11]. Hence, our observations about inequity are not unique to NFM; since the introduction of compulsory competitive tendering to England in the 1980s, this approach has encountered challenges related to unequal capacity for application and implementation, such as for urban greenspace initiatives[56]. Given the current global climate of widespread financial constraint, it is likely that market-based instruments will continue as popular approaches for the allocation of scarce resources to environmental projects, including NFM[50]. We encourage the NFM and wider research community to consider the unintended consequences of 'mainstreaming' NbS through competitive tenders; we are at risk of reinforcing persistent inequities[24,47]. Thus, we present some recommendations for equitable NFM resource allocation that may also be applicable to a broad suite of environmental projects in the UK and elsewhere that adopt a competitive tender process.

### Recommendations for equitable NbS and tender

These recommendations are intended to address the inequity demonstrated in the above analysis. However, the core messages are applicable to environmental projects funded through competitive tender, whereby the outcomes of resource allocation are likely to produce benefits for only some 'at need' groups and regions, risking creating or reinforcing inequity. The implications for tender design and delivery are synthesised in Fig. 4, which is intended to guide equitable resource allocation.

1) Barriers to applications must be proactively addressed, as already 'Enthusiastic communities' often enable tender application and success. Less fortunate communities and regions with equivalent need and suitability for NbS may be unintentionally excluded from tendered opportunities. A preliminary investigation is needed to identify communities typically under-represented in tender processes within environmental landscapes that would benefit from cost-effective NbS. Thus, the first step towards equitable tendering should involve scoping and outreach (Fig. 4) to ensure that eligible communities in more deprived areas are equally aware of tender opportunities and empowered to engage with competitive funding. Scoping and engagement for outreach should be undertaken with existing agencies, groups, not-for-profits and networks already working closely with vulnerable groups to ensure the transparency and trustworthiness of the tender process.

2) Our analysis revealed that urban deprived areas tended to be more successful than deprived rural areas. This may be linked to the accessibility of these areas to support the application process through existing support networks. Therefore, we suggest that funding streams require flexible options to facilitate the involvement or capacity building of groups, agencies and networks that often support vulnerable communities in other capacities, such as through employability

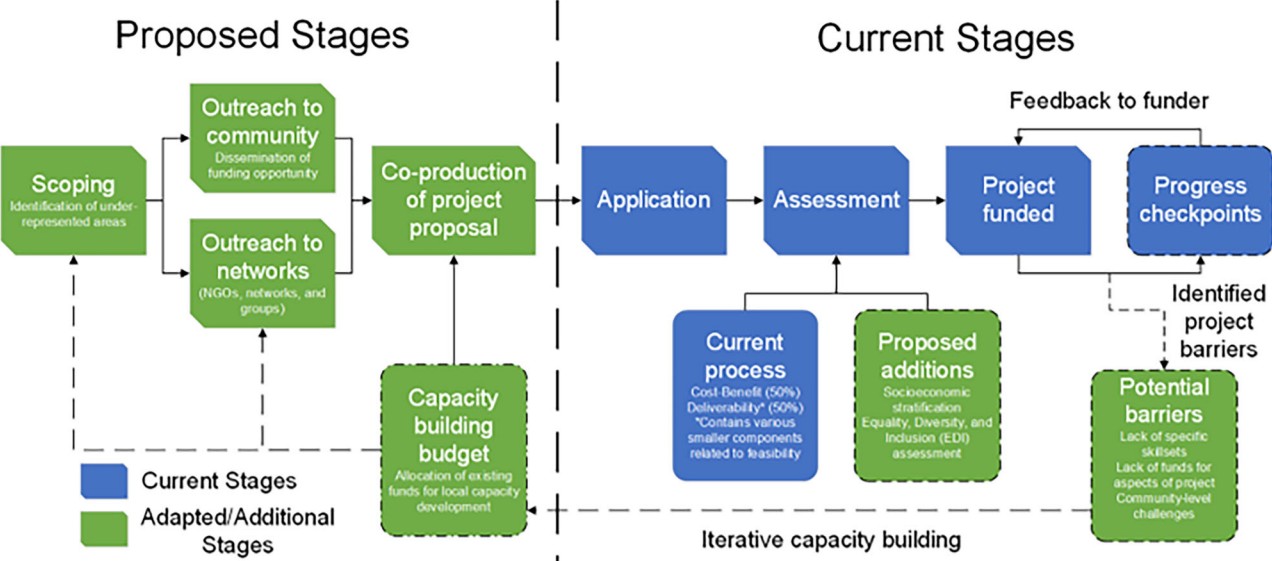

**Fig. 4 | A schematic showing the current and proposed stages of the NFM funding process for improving EDI best practice (Adapted from Ngai et al.[76]).** Proposed stages refers to additional stages that will need to be integrated into the application and assessment process for NFM funding to improve equitable outcomes. Current stages refers to existing stages that form part of the application and assessment process for NFM funding which may require adaptation to improve equitable outcomes. Green boxes highlight additional/adapted stages of the process, while Blue boxes highlight the existing stages. Arrows provide direction of flow in the diagram.

and soft-skill programmes, to co-develop tender applications; to encourage rural deprived communities to submit applications.

3) Funders should embed socioeconomic stratification into tender assessment processes to ensure equitable representation of communities from diverse backgrounds in resource allocation for NbS, similar to FCERM[37], to ensure that the projects funded are not biased to affluent areas, as our analysis suggests. However, this process should not preclude consideration of appropriate locality and feasibility for effective flood protection. The success of stratification may depend on the effectiveness of engaging peripheral groups and networks to support application development, with the aim to build enduring capacity within under-represented groups and regions.

4) Finally, we suggest that as part of the application assessment process, there should be a consideration of Equality, Diversity, & Inclusion (EDI). Much like the adoption of EDI best practice in the hiring process for the workforce internationally, the development of EDI best practice for environmental tendering will necessarily involve some capital investment, such as for supporting application development and project delivery within more deprived communities. This additional investment should not be viewed as a 'cost' in the traditional sense of cost-benefit decision-making. Rather, achieving equitable flood protection should be viewed as an essential criterion for gauging the overall success of NBS programmes. Similar approaches are evidenced through schemes such as the US EPA's Environmental Justice Small Block Grants, to better prioritise NFM funds allocated based on need rather than purely cost-benefit and feasibility[57,58].

Our proposed recommendations and approach (Fig. 4) are widely applicable to environmental projects funded through tender processes, such as habitat restoration, as the skills and capacity building required to manage and tender for these projects are cross-cutting. Lessons learned from the early days of NbS tendering could inform policy and practice towards avoiding past mistakes for future environmental projects.

## Methods

To assess the equitability of NFM resource allocation, we used GIS geospatial methods combined with hypothesis testing using a Monte Carlo sampling approach to first identify the rurality and deprivation characteristics of NFM application and funded project sites from the 2017 and 2023 NFM funding programmes and second, compare their statistical representativeness nationally.

Figure 5 outlines the approach taken for processing the GIS data, including: Data Inputs, GIS Operations and Data Cleaning and Outputs, which were then used in the statistical analysis. These three stages are explained in turn, after which the statistical testing methodology is outlined.

### Data inputs

The data used in this study correspond to two DEFRA funding programmes: The 2017 NFM programme that ran from 2017 to 2021, and the ongoing 2023 NFM programme, for which successful projects were announced in February 2024. For the 2017 programme, 60 pilot projects were funded, split into 26 catchment-scale projects led by risk management authorities and 34 community-scale projects (only 33 out of 34 were implemented, hence analysis of only 33 schemes). The former projects were based on pre-authorised flood risk authority schemes, which did not require an application process, hence their exclusion from our analysis. For the 2023 programme, all projects were part of the competitive tendering process and were therefore included in our analysis.

Data on funded projects from the 2017 and 2023 programmes were taken from publicly available programme reporting, press releases and technical documents, with 2023 project locations manually geo-positioned using ArcGIS Pro. Data on unsuccessful application project locations for both programmes were provided by DEFRA via a Freedom of Information request (FOI-NR366962). A full list of the data sources used is provided in Supplementary Table 1, with links to each source.

### Data cleaning and GIS operations

ArcGIS Pro version 2.6.0[59] was used to visualise and process all data. In the initial stage, data that either fell outside of the UK boundaries or could not be displayed due to invalid locational co-ordinates was removed along with duplicates, which was particularly important for the unsuccessful applications. The 2023 data for unsuccessful applications contained instances of multiple applications for the same project, either with identical or similar intervention locations. There were also some entries with no location data.

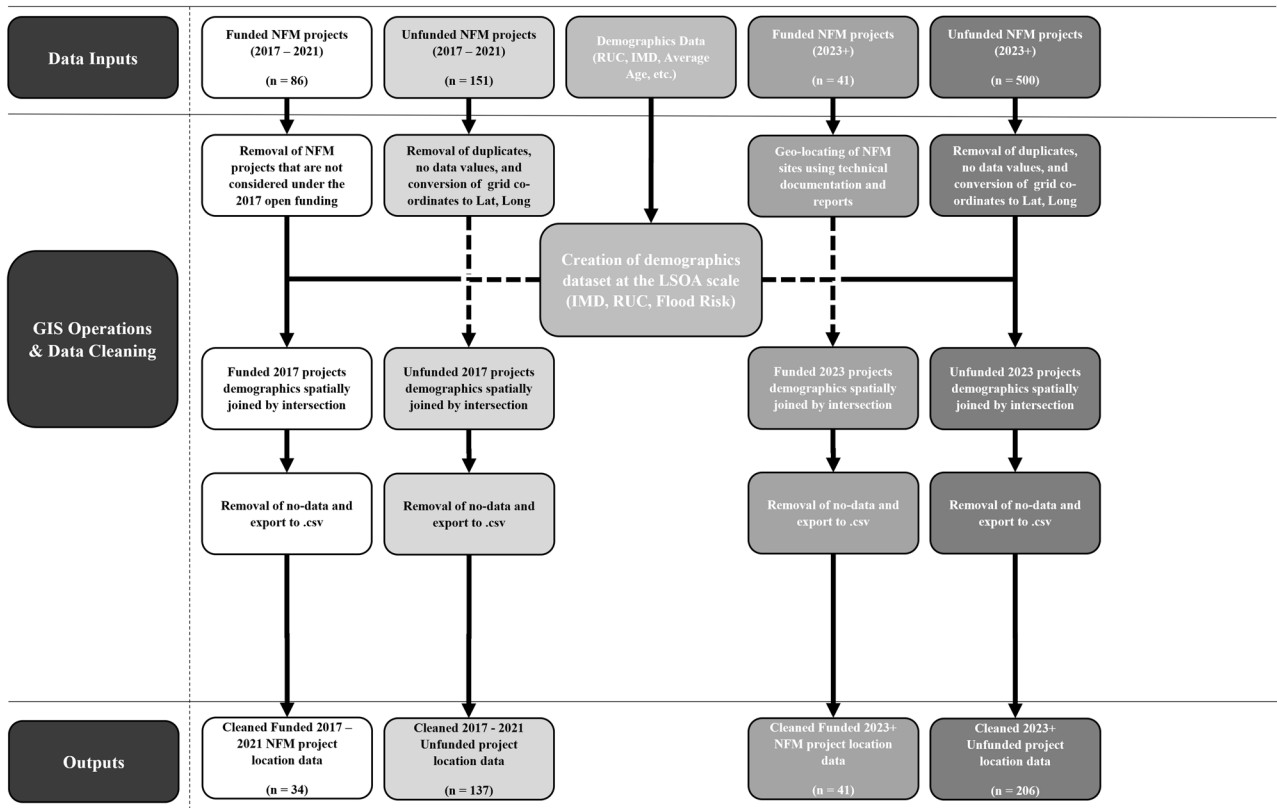

**Fig. 5 | A schematic of the GIS processing workflow highlighting the data sources, geospatial methods and cleaning approaches applied to produce the final application and project data.** Boxes are coloured from dark grey to white to represent different data outcomes and stages. The far left (darkest grey) represents the various stages of the schematic. All other boxes refer to either data, processes, or outputs that occur during these stages. Arrows provide direction of flow in the diagram.

The data provided did not group multiple applications into single projects. Therefore, individual applications were grouped into projects using nearest-neighbour clustering with a predefined distance threshold. Distance thresholds between 1000 and 10,000 m were applied, and the elbow method was used to determine the optimum threshold (4000 m, see Fig. 6a). It was assumed that applications missing location data were part of multiple-application projects and therefore these were ignored. Following manual examination of one borderline case (3990 m separation distance), the clustering process produced 206 clusters, matching the expected number of unsuccessful projects. The potential impact of this clustering on the analysis of 2023 applications is discussed further below.

The next stage involved the combination of the Rural Urban Classification[60], the UK Index of Multiple Deprivation[61] and UK FSV[34] data, at the LSOA level.

The IMD pertains to seven key domains of deprivation (Income, Employment, Education, Health, Crime, Barriers to Housing & Services, and Living Environment), and has been commonly used to identify inequities[32,62–64]. It is important to note that the IMD tends to under-represent rural poverty due to the spatial dispersion and access to green spaces in these areas[65,66]. However, when the IMD deciles are considered for rural areas only, we see that a majority of the NFM sites still fall directly within the least deprived areas, as shown in Fig. 2. The adjusted rural deprivation index (RDI)[67], shows this clearly, but is yet to be made public, and so our use of the RUC dataset to identify differences in deprivation between classifications was deemed acceptable. Burke and Jones[68] have also provided a potential RDI method for Norfolk, but again, it is yet to be made nationally available.

The RUC identifies and characterises settlements based on population densities and sparsity at varying scales, which is then split into eight separate categories: four urban and four rural[33,69]. The RUC has strong statistical grounds for this classification, and with little change between 2001 and 2011 RUC in terms of sparsity, we can assume that the 2011 classifications still hold true in 2024. By classifying NFM sites as either being in rural or urban areas, we can identify key differences between IMD distribution within urban areas and rural areas from both the applications and funded stages. The split between urban and rural funded projects for 2017 was $n = 18$ rural, and $n = 16$ urban; and for 2023, it was $n = 23$ rural, and $n = 18$ urban. This is particularly important as typically NFM is conducted in rural areas[70], and so the key story around urban NFM inequity may be lost without this distinction. Furthermore, drivers of inequity within urban areas can be vastly different from those of rural areas[71], and the levels of interest and knowledge around the environment may also be different[72].

The key GIS technique employed was the spatial intersection and common identifier join tools. A spatial join combines the attribute tables of two spatial layers based on the spatial relation between their geometries[73]. For this case, we used the intersection of points with the underlying LSOA polygon data. Intersection relies on the Dimensionally Extended 9-Intersection Model to perform this spatial relation, and overlays one dataset with another to find matrix intersection[74,75]. The join is based on common identifiers, considering a common identifier within the database file. As the RUC and IMD are both available at the LSOA scale, it is a common identifier between both datasets, and so was selected as the join field. For projects with multiple locations, a mean IMD value was calculated for each project group. For the 2023 applications, this was done automatically using the clustered data, providing some uncertainty regarding these groups. The average standard deviation in IMD deciles within groups compared to the rounded group mean was 0.86 deciles. Compared to the ungrouped intervention site data, this averaging leads to a reduction in certain deciles (particularly decile 10 for both programmes and decile 3 for urban projects), and an increase within more central deciles (Fig. 6b).

**Fig. 6 | Results of the clustering algorithm showing how the clustering distance used affected the number of clusters of NFM projects identified and compares the grouped and ungrouped to the national distribution. a** The impact of distance threshold on the number of clusters. **b** The distribution of all ungrouped intervention site data (dotted staircase) compared to the grouped projects (grey shading) and the national distribution (grey staircase).

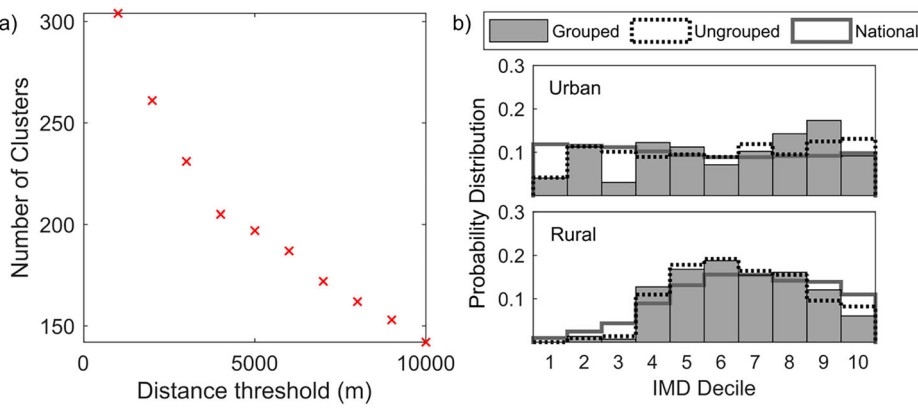

Deciles where this difference changed the statistical significance within the probability distributions have been labelled in Fig. 2.

### Non-parametric Monte Carlo sampling and hypothesis testing

To evaluate the representativeness of the NFM applications and funded projects compared to the national IMD distribution, two approaches were used. First, a one-sample chi-squared test was carried out to test the null hypothesis that the samples of applications and funded projects follow the same distribution as their populations (the applications and national distribution, respectively). In cases where the expected frequency for a decile was less than 5, IMD deciles were merged until the combined value exceeded 5. Due to the distinct difference in IMD profile nationally, urban and rural sites, classified as described above, were considered separately for all statistical analyses. A significance level of $\alpha = 0.05$ was used consistently across all analyses.

Second, Monte Carlo sampling of the national distribution and NFM application populations was undertaken. In each case, the underlying population was Monte Carlo sampled, without replacement, to produce $M = 1,000,000$ Monte Carlo samples ($X^j$), each with a sample size ($N$) equal to that of the observed sample.

Two different analyses were undertaken. First, the samples of funded projects ($x$) were compared to the national IMD distribution. Four test statistics ($T_{A-D}$) of the funded project samples were compared to their respective sampling distribution across all Monte Carlo samples. The four test statistics were the mean IMD decile ($T_A$), the minimum IMD decile ($T_B$), the percentage with IMD < 5 ($T_C$) and the percentage with IMD > 6 ($T_D$).

$$T_A(X^j) = \frac{1}{N}\sum_{i=1}^{N} X_i^j \tag{1}$$

$$T_B(X^j) = \min(X^j) \tag{2}$$

$$T_C(X^j) = \frac{100}{N}\sum_{i=1}^{N} I(X_i^j < 5) \tag{3}$$

$$T_D(X^j) = \frac{100}{N}\sum_{i=1}^{N} I(X_i^j > 6) \tag{4}$$

Here, $I(x)$ is the indicator function, equal to 1 if $x$ is true, or 0 otherwise. The mean was selected as a measure of central tendency, whereas the other three statistics were chosen to identify specific patterns within the distribution which may not be reflected in the mean (i.e. absence of low deciles, relative under-/over-representation of low/high deciles). One-tailed hypothesis testing was conducted using the percentile method for the four test statistics, in each case assuming the null hypothesis, $H_0$, that there was no bias against lower IMD deciles in the selection of funded projects. For each hypothesis test, $p$ values were calculated empirically using the percentile method such

that:

$$\begin{aligned} p_{A,B,D} &= P(T_{A,B,D}(X) \geq T_{A,B,D}(x)|H_0 \text{true}) \\ &= \frac{1}{M}\sum_{j=1}^{M} I(T_{A,B,D}(X^j) \geq T_{A,B,D}(x)) \end{aligned} \tag{5}$$

$$p_C = P(T_C(X) \leq T_C(x)|H_0 \text{true}) = \frac{1}{M}\sum_{j=1}^{M} I(T_C(X^j) \leq T_C(x)) \tag{6}$$

For the second analysis, a similar Monte Carlo approach was applied but application and funding stages were disaggregated. Therefore, the applications and funded projects ($x$) were compared to the Monte Carlo samples of the underlying national IMD distribution and applications, respectively ($X^j$), mirroring the chi-squared test procedure. The test statistic, $T_k(X^j)$, for each IMD decile $k$, was the proportion of data within that decile, and two-tailed hypothesis testing was undertaken under the null hypothesis that the proportion within that decile was representative of random selection from the underlying distribution. For each decile, $p$ values were calculated empirically using the percentile method.

$$T_k(X^j) = \frac{1}{N}\sum_{i=1}^{N} I(X_i^j = k) \tag{7}$$

$$p_k = \begin{cases} \frac{2}{M}\sum_{j=1}^{M} I(T_k(X^j) \geq T_k(x)) & T_k(x) > \widetilde{T_k(X^j)} \\ \frac{2}{M}\sum_{j=1}^{M} I(T_k(X^j) \leq T_k(x)) & T_k(x) \leq \widetilde{T_k(X^j)} \end{cases} \tag{8}$$

Here, the tilde symbol denotes the median of the Monte Carlo samples. The analysis was repeated using the ungrouped 2023 application data to examine any changes to the findings.

The representativeness of the applications and funded projects was also assessed in terms of flood risk and existing flood protection using one-sample chi-squared tests to test the null hypotheses that the samples of applications and funded projects follow the same distribution as their populations (the applications and national distribution, respectively).

### Limitations

There are some potential limitations of our study, such as: small study sample sizes, the manual geo-locating of 2023 funded projects, the use of the clustering algorithm for the 2023 unfunded projects and the fact that IMD under-represents rural deprivation. The small number of projects funded through the national NFM programme could explain the reason for this IMD skewing; however, our use of Monte Carlo sampling aims to highlight that even with this limitation, the statistical likelihood of the distribution is still significant. While the manual geo-locating of 2023 funded projects was conducted using a mix of technical reports available online, DEFRA sources, and other documentation, this could lead to inaccuracies in the real-world locations of the features. This is unavoidable due to a lack of data. However,

we do commonly find that there is a clustering of similar deciles spatially (Lloyd et al.[32]), and so we can assume that even given some locational discrepancies, we are likely to see similar IMD scores. As discussed above, the clustering of multiple locations into projects for the unsuccessful 2023 applications also introduces uncertainty. However, this only affects one subset of data and Fig. 6 shows that it does not have a substantial impact on the distributions. Nevertheless, we have clearly signified where this uncertainty could affect the statistical significance of our results. Finally, a common critique of the IMD is the under-representation of rural deprivation. Burke et al.[68] have been developing an RDI that shows a 1–2 decile change for areas when compared to the IMD; however, without access to these data, it was not possible to account for this. Furthermore, as there is no Urban Deprivation Index, it would not be possible to use this for this study. Future considerations of rural deprivation dynamics should be considered to help improve the accuracy of the results presented here.

## Reporting summary

Further information on research design is available in the Nature Portfolio Reporting Summary linked to this article.

## Data availability

All data sources can be found in the Supplementary Table 1 in the Supplementary materials; however, the datasets created and analysed to create the figures shown can be found via the following GitHub Repository link: https://github.com/BartHill/Market-based-instruments-to-fund-nature-based-solutions-Data.git.

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

## Acknowledgements
This research is supported by the EPSRC Centre for Doctoral Training in Water and Waste Infrastructure Systems Engineered for Resilience (Water-WISER; EP/S022066/1). We also thank the UK Environment Agency for providing the data related to NFM project applications through the 'Freedom of Information' request, NR366962, which made this research possible and is available through an Open Government Licence.

## Author contributions
B.H., T.M. and H.M. conceived and designed the initial study, with B.H. and T.M. conducting the geospatial and geostatistical analysis of the data. B.H. wrote the initial manuscript, with T.M. co-designing many of the figures and results. H.M., L.B. and M.G. all supported the writing, reviewing and editing of the manuscript over successive iterations, with L.B. and M.G. providing supervision and mentorship to B.H.

## Competing interests
The authors declare no competing interests.
