## [Transparent Peer Review file · Communications Earth & Environment]

Market-based instruments to fund nature-based solutions for flood risk management can disproportionately benefit affluent areas.

Corresponding Author: Dr Bartholomew Hill

Version 0:

Decision Letter:

Dear Mr Hill,

Your manuscript titled "Nature-based solutions for flood risk management: is competitive funding an equitable approach?" has now been seen by 3 reviewers, whose comments are appended below. You will see that they find your work of some potential interest. However, they have raised quite substantial concerns that must be addressed. In light of these comments, we cannot accept the manuscript for publication, but would be interested in considering a revised version that fully addresses these serious concerns.

We hope you will find the reviewers' comments (attached below) useful as you decide how to proceed. Should additional work allow you to

- address these criticisms (that is, either to incorporate the suggestions or provide a compelling argument why the point made by the reviewer is not valid, or relevant to the editorial threshold as outlined below)

AND

- meet our editorial thresholds as outlined below,

then we would be happy to look at a substantially revised manuscript.

Editorial thresholds:

- 1) Provide novel and firmly supportive insight into the inequity in funding application and distribution for Nature-based Solutions in flood risk management in England.
- 2) Outline your method and data in detail, including the statistical approach, explain deciles and significance level, clarify all terms used, and use them consistently in the text.
- 3) Expand your discussion of your findings in regard to relevance for wider Nature Based-Solution implementation and ecological restoration and demonstrate that all your data and analysis fully support your claims.

When resubmitting, please provide a point-by-point response to the reviewers' comments. Please submit your responses as a separate file, distinct from your cover letter where you can add responses to the Editors' comments that you do not want to be made available to the reviewers. Word files are preferred. We recommend that any figures, tables or graphs that are included in the response to reviewers are also included in the main article or Supplementary Information.

If the revision process takes significantly longer than three months, we will be happy to reconsider your paper at a later date, as long as nothing similar has been accepted for publication at Communications Earth & Environment or published

elsewhere in the meantime.

Please use the following link to submit your revised manuscript, point-by-point response to the reviewers' comments with a list of your changes to the manuscript text (which should be in a separate document to any cover letter), a tracked-changes version of the manuscript (as a PDF file) and any completed checklist:

Link Redacted

Please do not hesitate to contact us if you have any questions or would like to discuss the required revisions further. Thank you for the opportunity to review your work.

Best regards,

Sisi Meng
External Editor
Communications Earth & Environment

Martina Grecequet, PhD
Senior Editor
Communications Earth & Environment

EDITORIAL POLICIES AND FORMAT

If you decide to resubmit your paper, please ensure that your manuscript complies with our editorial policies and complete and upload the checklist below as a Related Manuscript file type with the revised article:

Editorial Policy Policy requirements
(Download the link to your computer as a PDF.)

- Behavioural and social science
- Ecological, evolutionary & environmental sciences
- Life sciences

<https://www.nature.com/documents/nr-reporting-summary.zip>

For your information, you can find some guidance regarding format requirements summarized on the following checklist: (<https://www.nature.com/documents/commsj-phys-style-formatting-checklist-article.pdf>) and formatting guide (<https://www.nature.com/documents/commsj-phys-style-formatting-guide-accept.pdf>).

REVIEWER COMMENTS:

Reviewer #1 (Remarks to the Author):

This manuscript examines the equity of allocating resources for natural flood management in the UK in the recent past through two DEFRA funding schemes, drawing from quantitative analysis of data combining project details and deprivation and other socio-economic indicators. The manuscript reports results that are both new and important and it has clear potential to make a contribution to the literature on equity and social justice aspects of funding and implementing nature based solutions.

The manuscript only needs very minor revisions before it can be accepted for publication.

First, the authors could define and explain better "competitive tenders" as a policy instrument very early in the manuscript, as it is rather central term for the work and yet is currently not explained. Not all readers will have intuitive understanding of their key features.

Second, the authors observe that significant proportion of rural NFM projects were in protected areas or other special places.

The explanation likely is that siting NFM projects is easier on publicly or institutionally owned land. This is a finding of Garvey and Paavola (2021) whose other findings resonate also with authors' arguments about the importance of access to expertise and experience as factors behind less deprived areas' success in bidding for funding for NFM.

Third, while the discussion articulates the findings well with regard to research allocation for NFM in the UK and its equity dimensions, the authors should make more effort to articulate the contribution of their article to the wider NBS implementation and ecological restoration literature.

Reference

Garvey A, Paavola J (2021) Community action on natural flood management: governing a catchment-based approach in the UK. *Environmental Policy & Governance* 32: pp. 3-16. <https://onlinelibrary.wiley.com/doi/full/10.1002/eet.1955>

Reviewer #2 (Remarks to the Author):

This study addresses the relatively neglected question whether NBS funding resources have been equitably distributed, with an assumption that there is a likelihood that the competitive NBS funding allocation can exacerbate existing inequalities. The authors explore two recent competitive NFM funding schemes by DEFRA, providing statistical evidence of the extent to which they distribute resources for NFM and associated co-benefits equally. Finding that NFM resources are invested more in affluent areas—applications in more deprived areas are less likely to be successful compared to applications for projects in more affluent areas and few applications were submitted to NFM programmes for projects in more deprived areas compared to more affluent areas, this research suggests an inequality. It also suggests that deprived areas with high flood risk receive less funding for NBS projects than their more affluent counterparts. The article raises awareness of the potential problem of using competitive tendering as a way to fund NBS projects. This article is well-written, clearly accounting for the methods and delivering the messages, and I struggle to find faults. I only have two minor suggestions to for the authors to consider to enrich this article:

1) In Discussion, the authors focus on the implication of the IMD analysis, only touched upon flood risk implicitly. While the authors do state that deprived areas with high flood risk receive less funding, there should be more detailed, explicit discussion on whether DEFRA's competitive tendering addresses areas with higher flood risks.

2) Towards the end of the article the authors mention "...many NFM projects are funded by local communities and charities..." (line 239-240) as one of the limitations that their findings may not represent all cases. This is fair, but it would be helpful if the authors can provide facts (e.g., relevant statistics) to state this already in Introduction such that the readers have a sense of the degree to which NBS (or NFM) funding mechanism is through competitive tendering.

Reviewer #3 (Remarks to the Author):

The authors are proposing that competitive tenders for Nature-based Solutions in flood risk management do not well account for inequalities at both application and funding stages, which potentially undermines the equality benefits these solutions aim to provide. They are analysing two national flood management programs in England to demonstrate how market-based funding instruments can create socioeconomic and geographical disparities in resource allocation, challenging the effectiveness of competitive funding approaches.

Specific comments

I comment the authors for the very interesting and relevant research. I enjoyed reading the manuscript, however I recommend that the following comments should be addressed before considering publication at *Nature Communications Earth & Environment*.

The discussion and usage of the words of inequality and equity could be clarified to ensure consistency in terminology.

Currently, the terms "inequality" and "equity" appear to be used interchangeably, which causes some confusion.

- Line 10: The abstract does not clearly define what is meant by "inequality", which is given the context, helpful to specify whether the focus is on disparities in access to funding (inequality) or for example whether the allocation process ensures fair distribution (equity).

- Line 66: The term "equality" may not be the most accurate choice here. Since the discussion centres on the distribution of funds, "equity" would be a more precise term to reflect the goal of fair allocation rather than equal treatment.

The authors provide a valuable analysis of equity issues in competitive funding for Nature-based Solutions in flood management, however to strengthen the paper's impact for practitioners and policymakers, I suggest developing a dedicated 'Lessons Learned' section.

- The section on 'Lessons Learned' may explicitly synthesize the observations stated into transferable recommendations.

- The authors could propose or outline a framework or decision tree for policymakers to benefit from evaluation and improve equity in such competitive funding approaches.

- Providing specific examples of successful alternative funding models that have achieved better equity outcomes if there are any.

- Give more details or a roadmap on how to implement the suggestions like "stratifying the tender process" or "providing support for less experienced communities."

• This section may as well expand on the transferability of the findings to other environmental management contexts beyond flood management.

Line 36f: NbS is introduced and linked to inequality, however it would be good to provide some examples in the text here on how exactly it addressing social and well-being problems. How is well-being linked to reduce inequality?

In the analysis is population normalized? Is the population for floodplains only, if so are protection levels included?

Line 190 how does it promote equitable flood protection? Also no references included here.

Technical comments

Line 40: NBS should be NbS, check throughout.

Line 91: how significant?

At results sections, it may benefit from providing an introduction to what these deciles represent, like 1 lowest deprived etc.

Communications Earth & Environment is committed to improving transparency in authorship. As part of our efforts in this direction, we are now requesting that all authors identified as 'corresponding author' create and link their Open Researcher and Contributor Identifier (ORCID) with their account on the Manuscript Tracking System prior to acceptance. ORCID helps the scientific community achieve unambiguous attribution of all scholarly contributions. You can create and link your ORCID from the home page of the Manuscript Tracking System by clicking on 'Modify my Springer Nature account' and following the instructions in the link below. Please also inform all co-authors that they can add their ORCIDs to their accounts and that they must do so prior to acceptance.

Version 1:

Decision Letter:

Dear Mr Hill,

Your manuscript titled "Nature-based solutions for flood risk management: is competitive funding an equitable approach?" has now been seen by our reviewers, whose comments appear below. In light of their advice we are delighted to say that we are happy, in principle, to publish a suitably revised version in Communications Earth & Environment.

We therefore invite you to revise your paper one last time to address the remaining concerns of our reviewers. In particular, for publication in Communications Earth & Environment we request that you provide a discussion of funding patterns between protected and unprotected areas, as well as the impact on inequality findings when accounting for existing infrastructure.

At the same time we ask that you edit your manuscript to comply with our format requirements and to maximise the accessibility and therefore the impact of your work.

EDITORIAL REQUESTS:

****Please take care to match our formatting and policy requirements. We will check revised manuscript and return manuscripts that do not comply. Such requests will lead to delays. ****

SUBMISSION INFORMATION:

OPEN ACCESS:

Communications Earth & Environment is a fully open access journal. Articles are made freely accessible on publication. For further information about article processing charges, open access funding, and advice and support from Nature Research, please visit <https://www.nature.com/commsenv/open-access>

Link Redacted

Best regards,

Sisi Meng, PhD
Editorial Board Member
Communications Earth & Environment

Martina Grecequet, PhD
Senior Editor,
Communications Earth & Environment
Consulting Editor
Communications Sustainability

REVIEWERS' COMMENTS:

Reviewer #1 (Remarks to the Author):

The authors have satisfactorily addressed all of my comments on the original version in their revised manuscript version, which is clearly strengthened and can now be accepted for publication.

Reviewer #2 (Remarks to the Author):

I think the authors have addressed all my comments.

Reviewer #3 (Remarks to the Author):

The authors have done very well in responding to the referee comments and improved a lot. Now the manuscript reads well and reflects the obvious good quality of the work that was done. I enjoyed reading the "Recommendations for equitable NbS and tender" section that the authors have included based on earlier suggestions. After the authors have responded properly to my final comment, I would suggest to consider this manuscript for publication at Nature Communications Earth & Environmental.

I maintain that protection levels may be or are a critical confounding factor that should be addressed. Even the binary YES/NO protection data you mention could influence your equity findings. It is also the case that most protection levels are for specific areas as there will be a flood design to mitigate risks in larger areas. Areas with existing protection may appear lower priority for funding which is potentially masking whether funding disparities reflect genuine need differences or historical investment patterns. Could you at least conduct a sensitivity analysis comparing funding patterns between protected and unprotected LSOAs to test whether your inequality findings hold when accounting for existing infrastructure? Alternatively, this should be mentioned and discussed in the manuscript clearly.

We thank both the Editor, and the three reviewers for their positive and constructive comments on this article which have improved the piece significantly. We have provided a detailed response in blue to each of the comments and the editorial thresholds mentioned.

Editorial thresholds:

1) Provide novel and firmly supportive insight into the inequity in funding application and distribution for Nature-based Solutions in flood risk management in England.

We are pleased that all three reviewers recognised the research as “new and important” (Rev. 1), “well-written, clearly accounting for the methods and delivering the messages” (Rev. 2) and “very interesting and relevant (Rev. 3). In response to the suggestion of Reviewer 3, we have further developed the “Lessons learned” section as recommendations and a schematic that has been worked into the final section of the article. This provides actionable methods for reducing inequity at the pre-application, application, and assessment phases. We have also clarified in the text that the terminology is focussed on the inequitable allocation of funding, and the impacts that may have in exacerbating existing inequities within communities.

2) Outline your method and data in detail, including the statistical approach, explain deciles and significance level, clarify all terms used, and use them consistently in the text.

The statistical methods have now been thoroughly introduced in the methods section, including additional Chi squared tests and formal setting out of the percentile method hypothesis testing. Significance level has now been standardised at 0.05 across all tests and figures and text edited to reflect this. The deciles have been explained, in response to Reviewer 3’s comment, and clarity around Figure 3 has been provided. Terminology (e.g. “NbS”, “equity/equality”) is now used consistently.

3) Expand your discussion of your findings in regard to relevance for wider Nature Based-Solution implementation and ecological restoration and demonstrate that all your data and analysis fully support your claims.

Alongside the “Recommendations” section added to the article, we have adapted the discussion to be relevant to the wider audience beyond just NFM or NbS to the wider challenges around the allocation of funding for environmental projects. Furthermore, we draw upon examples of more equitable funding approaches taken by other sectors that could be utilised by NbS/NFM practitioners instead of the current approach. To ensure that all our claims are well-supported by our data and analysis, we have removed/rephrased one ambiguous statement identified by Reviewer 2 regarding public funding and have also added an additional Chi squared test to further support our findings.

REVIEWER COMMENTS:

Reviewer #1 (Remarks to the Author):

This manuscript examines the equity of allocating resources for natural flood management in

the UK in the recent past through two DEFRA funding schemes, drawing from quantitative analysis of data combining project details and deprivation and other socio-economic indicators. The manuscript reports results that are both new and important and it has clear potential to make a contribution to the literature on equity and social justice aspects of funding and implementing nature-based solutions.

We thank Reviewer #1 for their acknowledgement of the novelty and importance of our work.

The manuscript only needs very minor revisions before it can be accepted for publication.

We thank Reviewer #1 for their two minor comments, responses to which are provided below.

First, the authors could define and explain better "competitive tenders" as a policy instrument very early in the manuscript, as it is rather central term for the work and yet is currently not explained. Not all readers will have intuitive understanding of their key features.

We thank Reviewer #1 for raising this need for definition, we have now added the following sentence:

““Competitive tender” allows open bidding for project funding where the most economically competitive (e.g., greatest cost-benefit ratio or lowest cost) tender option is selected.”

Second, the authors observe that significant proportion of rural NFM projects were in protected areas or other special places. The explanation likely is that siting NFM projects is easier to publicly or institutionally owned land. This is a finding of Garvey and Paavola (2021) whose other findings resonate also with authors' arguments about the importance of access to expertise and experience as factors behind less deprived areas' success in bidding for funding for NFM.

We thank Reviewer #1 for their recommendation, and have added the following statement;

“A likely reason for the high representation of NFM in rural areas is the associated ease of implementation on publicly or institutionally owned land⁵¹.”

We have also discussed in more detail around how private funding being scarce or nascent is another potential reason for this targeting of funding.

Third, while the discussion articulates the findings well with regard to research allocation for NFM in the UK and its equity dimensions, the authors should make more effort to articulate the contribution of their article to the wider NBS implementation and ecological restoration literature.

Throughout the article, we have now added further details on how this process could be applied to other tender for the environment schemes and have added reference to the US EPAs Environmental Justice Small Block Grants that could provide both lessons learned and a way of operationalising the proposed recommendations.

Reference

Garvey A, Paavola J (2021) Community action on natural flood management: governing a catchment-based approach in the UK. *Environmental Policy & Governance* 32: pp. 3-16. <https://onlinelibrary.wiley.com/doi/full/10.1002/eet.1955>

We thank Reviewer #1 for highlighting this piece of literature, we have now included this within the article.

Reviewer #2 (Remarks to the Author):

This study addresses the relatively neglected question whether NBS funding resources have been equitably distributed, with an assumption that there is a likelihood that the competitive NBS funding allocation can exacerbate existing inequalities. The authors explore two recent competitive NFM funding schemes by DEFRA, providing statistical evidence of the extent to which they distribute resources for NFM and associated co-benefits equally. Finding that NFM resources are invested more in affluent areas—applications in more deprived areas are less likely to be successful compared to applications for projects in more affluent areas and few applications were submitted to NFM programmes for projects in more deprived areas compared to more affluent areas, this research suggests an inequality. It also suggests that deprived areas with high flood risk receive less funding for NBS projects than their more affluent counterparts. The article raises awareness of the potential problem of using competitive tendering as a way to fund NBS projects. This article is well-written, clearly accounting for the methods and delivering the messages, and I struggle to find faults. I only have two minor suggestions to for the authors to consider to enrich this article:

We thank Reviewer #2 for their acknowledgement of the neglected question being addressed and appreciate their kind comment of “*struggling to find faults*”, as well as the two minor comments provided.

1) In Discussion, the authors focus on the implication of the IMD analysis, only touched upon flood risk inexplicitly. While the authors do state that deprived areas with high flood risk receive less funding, there should be more detailed, explicit discussion on whether DEFRA’s competitive tendering addresses areas with higher flood risks.

We thank Reviewer #2 for this comment, and in response, we firstly draw the reviewer’s attention to the statement in 154 – 156...

“These data show that variability in underlying flood risk between deciles does not explain the trends shown in Figures 1 and 2.”

We have now added further clarity within the discussion stating that...

“Furthermore, in both the application and funding stages we found no relationship between flood risk severity and NFM project siting, as well as fewer high flood risk areas within more

affluent deciles and so can dismiss this as a potential explanation for the targeting of NFM within certain areas. This finding reflects NFM's suitability to tackle smaller scale "nuisance" flooding rather than areas of very high flood risk⁵⁰."

A reason for our only brief touch upon flood risk, is due to the fact that we found no trend or statistical significance to the severity of flood risk being a predictor of NFM intervention targeting.

2) Towards the end of the article the authors mention "...many NFM projects are funded by local communities and charities..." (line 239-240) as one of the limitation that their findings may not represent all cases. This is fair, but it would be helpful if the authors can provide facts (e.g., relevant statistics) to state this already in Introduction such that the readers have a sense of the degree to which NBS (or NFM) funding mechanism is through competitive tendering.

We thank Reviewer #2 for highlighting this point. While no readily available statistic can be found for funding delivered through competitive tendering, we do know that 82% of NbS funding is public money and that competitive tendering is a widely used method for allocating public funds to NbS projects. We agree that our statement that "*many NFM projects are funded by local communities and charities*" lacked clarity.

We have addressed the reviewer's comment by (1) providing more qualitative detail about the popularity of competitive tendering within NbS funding and use of public funds for environmental management more broadly and (2) removed the later sentence within the discussion as part of a wider reworking of that section.

Reviewer #3 (Remarks to the Author):

The authors are proposing that competitive tenders for Nature-based Solutions in flood risk management do not well account for inequalities at both application and funding stages, which potentially undermines the equality benefits these solutions aim to provide. They are analysing two national flood management programs in England to demonstrate how market-based funding instruments can create socioeconomic and geographical disparities in resource allocation, challenging the effectiveness of competitive funding approaches.

We thank Reviewer #3 for their clear understanding of the findings of the article.

Specific comments

I comment the authors for the very interesting and relevant research. I enjoyed reading the manuscript, however I recommend that the following comments should be addressed before considering publication at Nature Communications Earth & Environment.

We thank Reviewer #3 for their interest and enjoyment from reading the article, as well as the comments provided.

The discussion and usage of the words of inequality and equity could be clarified to ensure consistency in terminology. Currently, the terms "inequality" and "equity" appear to be used interchangeably, which causes some confusion.

We thank Reviewer #3 for this comment, and have addressed this confusion between “*Inequality*”, “*Equality*”, “*Equity*”, and “*Inequity*”. We have opted to use the terms “*Equity*” and “*Inequity*”. Further information that relates to the specific line points below around terminology has been addressed.

- Line 10: The abstract does not clearly define what is meant by "inequality, which is given the context, helpful to specify whether the focus is on disparities in access to funding (inequality) or for example whether the allocation process ensures fair distribution (equity).

We thank you for identifying the need to define inequality and equity within context. We refer to “*equity*” as providing proportionally appropriate and fair allocations of funding. Hence there is an “*inequitable*” allocation of funds, which leads to an exacerbation of existing “*inequities*”.

- Line 66: The term "equality" may not be the most accurate choice here. Since the discussion centres on the distribution of funds, "equity" would be a more precise term to reflect the goal of fair allocation rather than equal treatment.

We have opted for the term “*equity*” instead of “*equality*”, reflecting the reviewer’s comment on the term.

The authors provide a valuable analysis of equity issues in competitive funding for Nature-based Solutions in flood management, however, to strengthen the paper's impact for practitioners and policymakers, I suggest developing a dedicated ‘Lessons Learned’ section:

- section on ‘Lessons Learned’ may explicitly synthesize on the observations stated into transferable recommendations. The
- The authors could propose or outline a framework or decision tree for policymakers to benefit from evaluation and improve equity in such competitive funding approaches.
- Providing specific examples of successful alternative funding models that have achieved better equity outcomes if there are any.
- Give more details or a roadmap on how to implement the suggestions like "stratifying the tender process" or "providing support for less experienced communities."

This section may as well expand on the transferability of the findings to other environmental management contexts beyond flood management.

We thank Reviewer #3 for this detailed recommendation. To operationalise this, we have decided to create five key recommendations, as well as a schematic for how to reduce inequity at the various stages of the NFM/NbS delivery process. This has added value to the article, and we believe our addition addresses this concern. We have also added a summary

statement of how policymakers and practitioners could learn from other exemplar funding streams that consider equity.

Line 36f: NbS is introduced and linked to inequality, however it would be good to provide some examples in the text here on how exactly it addressing social and well-being problems. How is well-being linked to reduce inequality?

We thank Reviewer #3 for this comment. We have added the statement on the following line

“In some cases, there are direct links to wellbeing such as restored wetland providing mental health benefits, however; reducing inequity associated with NFM/NbS implementation has the potential for much wider change (e.g., social mobility, development of soft and technical skills)”

In the analysis is population normalized? Is the population for floodplains only, if so are protection levels included?

In response to this, population is normalised by area. The Lower Super Output Area (LSOA) is a UK common geospatial area classification for a population between 1,000 and 3,000 people. Unfortunately, we cannot describe below this for “floodplains” only as the analysis described covers a national scale, and the data for populations at risk on flood plains is not available. Furthermore, not all communities at risk of flooding live on floodplains, and some can be affected within headwaters or coastal areas, hence our use of LSOA normalised by population. Finally, the dataset utilised does provided some detail as to “YES or NO” whether flood protection is provided per LSOA. However, this is not spatially described beyond the LSOA area and so may not give specific locations of where protection is provided.

Line 190 how does it promote equitable flood protection? Also no references included here.

We thank reviewer #3 for their comment and have added some further details including an additional reference.

“promoting equitable flood protection through equitable fund allocations, as well as profound wider benefits to community such as; soft and technical skills development, community empowerment, and social mobility”

Technical comments

Line 40: NBS should be NbS, check throughout.

We thank Reviewer #3 for identifying this oversight, and we have ensured that NbS is used throughout.

Line 91: how significant?

We agree that the reporting of significance lack clarity. We have now reported all p-values and used a consistent significance level of 0.05 throughout the text.

“There is statistically significant bias ($p < 0.05$) against lower IMD deciles in allocation of funded projects compared to the national IMD distribution”

At results sections, it may benefit from providing an introduction to what these deciles represent, like 1 lowest deprived etc.

We thank Reviewer #3 for noting this oversight, and have added the following statement to the introduction as well as a reminder at the beginning of the results for clarity...

“we use the UK Indices of Deprivation (IMD), an index used for assessing socioeconomic deprivation for small areas (~1500 people) where low deciles 1 – 4 represent less affluent areas, and 5 – 10 represent more affluent areas²⁸”

We thank both the Editor, and the three reviewers for their responses to our last revision of the manuscript. We have provided some comments in response below.

Editor:

Your manuscript titled "Nature-based solutions for flood risk management: is competitive funding an equitable approach?" has now been seen by our reviewers, whose comments appear below. In light of their advice we are delighted to say that we are happy, in principle, to publish a suitably revised version in Communications Earth & Environment.

We thank the editor for this response and welcome the opportunity to improve upon the manuscript.

We therefore invite you to revise your paper one last time to address the remaining concerns of our reviewers. In particular, for publication in Communications Earth & Environment we request that you provide a discussion of funding patterns between protected and unprotected areas, as well as the impact on inequality findings when accounting for existing infrastructure.

We have now provided an additional table that provides the outputs of an analysis of whether the LSOA's had existing flood protection or no and found that this had no influence on equity or allocation of funds. Instead, this represented similar rates to background national rates of flood protection. There was also no statistical significance in this as a predictor of NFM project membership.

At the same time we ask that you edit your manuscript to comply with our format requirements and to maximise the accessibility and therefore the impact of your work.

We have now reformatted our manuscript to comply with the Editorial Requests Table guidance.

All responses have been provided in the right hand column to each point raised on the document and is attached as a "Related Manuscript File".

Reviewer #1:

The authors have satisfactorily addressed all of my comments on the original version in their revised manuscript version, which is clearly strengthened and can now be accepted for publication.

We thank Reviewer #1 for this comment.

Reviewer #2:

I think the authors have addressed all my comments.

We thank Reviewer #2 for this comment.

Reviewer #3:

The authors have done very well in responding to the referee comments and improved a lot. Now the manuscript reads well and reflects the obvious good quality of the work that was done. I enjoyed reading the “Recommendations for equitable NbS and tender” section that the authors have included based on earlier suggestions. After the authors have responded properly to my final comment, I would suggest to consider this manuscript for publication at Nature Communications Earth & Environmental.

We thank Reviewer #3 for their previous review comments that helped to improve the manuscript, as well as their support for publication.

I maintain that protection levels may be or are a critical confounding factor that should be addressed. Even the binary YES/NO protection data you mention could influence your equity findings. It is also the case that most protection levels are for specific areas as there will be a flood design to mitigate risks in larger areas. Areas with existing protection may appear lower priority for funding which is potentially masking whether funding disparities reflect genuine need differences or historical investment patterns. Could you at least conduct a sensitivity analysis comparing funding patterns between protected and unprotected LSOAs to test whether your inequality findings hold when accounting for existing infrastructure? Alternatively, this should be mentioned and discussed in the manuscript clearly.

While the previous revised manuscript did provide evidence that flood risk was not a factor in determining NFM siting, we have now performed an additional analysis of the LSOA's that received NFM funding, had NFM applications, and national rates, to identify whether this was a predictor of NFM siting/funding. We found that there was no statistically significant relationship between whether an LSOA had flood protection or not and therefore is not a predictor of NFM funding. We have included this as a table within the manuscript, as well as some additional lines explaining this within the discussion section. This was based on the existing dataset that identifies LSOA flood risk that also detailed flood protection, and so the data was already joined and could be easily analysed for adding to the manuscript. We thank Reviewer #3 for highlighting this further as it only strengthens the findings of the article.